# Allyl Isothiocianate Induces Ca^2+^ Signals and Nitric Oxide Release by Inducing Reactive Oxygen Species Production in the Human Cerebrovascular Endothelial Cell Line hCMEC/D3

**DOI:** 10.3390/cells12131732

**Published:** 2023-06-27

**Authors:** Roberto Berra-Romani, Valentina Brunetti, Giorgia Pellavio, Teresa Soda, Umberto Laforenza, Giorgia Scarpellino, Francesco Moccia

**Affiliations:** 1Department of Biomedicine, School of Medicine, Benemérita Universidad Autónoma de Puebla, Puebla 72410, Mexico; rberra001@hotmail.com; 2Laboratory of General Physiology, Department of Biology and Biotechnology “L. Spallanzani”, University of Pavia, 27100 Pavia, Italy; valentina.brunetti01@universitadipavia.it (V.B.); giorgia.scarpellino@unipv.it (G.S.); 3Department of Molecular Medicine, University of Pavia, 27100 Pavia, Italy; giorgia.pellavio@unipv.it (G.P.); lumberto@unipv.it (U.L.); 4Department of Health Sciences, University of Magna Graecia, 88100 Catanzaro, Italy; teresa.soda@unicz.it

**Keywords:** allyl isothiocianate, hCMEC/D3, reactive oxygen species, Ca^2+^ signalling, nitric oxide, plasma membrane Ca^2+^-ATPase, store-operated Ca^2+^ entry

## Abstract

Nitric oxide (NO) represents a crucial mediator to regulate cerebral blood flow (CBF) in the human brain both under basal conditions and in response to somatosensory stimulation. An increase in intracellular Ca^2+^ concentrations ([Ca^2+^]_i_) stimulates the endothelial NO synthase to produce NO in human cerebrovascular endothelial cells. Therefore, targeting the endothelial ion channel machinery could represent a promising strategy to rescue endothelial NO signalling in traumatic brain injury and neurodegenerative disorders. Allyl isothiocyanate (AITC), a major active constituent of cruciferous vegetables, was found to increase CBF in non-human preclinical models, but it is still unknown whether it stimulates NO release in human brain capillary endothelial cells. In the present investigation, we showed that AITC evoked a Ca^2+^-dependent NO release in the human cerebrovascular endothelial cell line, hCMEC/D3. The Ca^2+^ response to AITC was shaped by both intra- and extracellular Ca^2+^ sources, although it was insensitive to the pharmacological blockade of transient receptor potential ankyrin 1, which is regarded to be among the main molecular targets of AITC. In accord, AITC failed to induce transmembrane currents or to elicit membrane hyperpolarization, although NS309, a selective opener of the small- and intermediate-conductance Ca^2+^-activated K^+^ channels, induced a significant membrane hyperpolarization. The AITC-evoked Ca^2+^ signal was triggered by the production of cytosolic, but not mitochondrial, reactive oxygen species (ROS), and was supported by store-operated Ca^2+^ entry (SOCE). Conversely, the Ca^2+^ response to AITC did not require Ca^2+^ mobilization from the endoplasmic reticulum, lysosomes or mitochondria. However, pharmacological manipulation revealed that AITC-dependent ROS generation inhibited plasma membrane Ca^2+^-ATPase (PMCA) activity, thereby attenuating Ca^2+^ removal across the plasma membrane and resulting in a sustained increase in [Ca^2+^]_i_. In accord, the AITC-evoked NO release was driven by ROS generation and required ROS-dependent inhibition of PMCA activity. These data suggest that AITC could be exploited to restore NO signalling and restore CBF in brain disorders that feature neurovascular dysfunction.

## 1. Introduction

The central nervous system (CNS) is not able to store energy, and therefore, neuronal activity is maintained by the continuous supply of oxygen and nutrients through the capillary bed [1], which accounts for ~90% of brain vasculature [2]. The mechanisms by which an increase in the firing rate leads to an increase in cerebral blood flow (CBF) to active neurons is known as neurovascular coupling (NVC) or functional hyperemia [1]. Capillary endothelial cells (cECs), which represent the core component of the blood–brain barrier (BBB), are also able to sense neuronal activity and thereby signal upstream arterioles to dilate and increase CBF to firing neurons [3,4,5,6]. In mouse brain cortical vasculature, synaptically released neurotransmitters cause an increase in endothelial Ca^2+^ concentration by stimulating inositol-1,4,5-trisphosphate (InsP_3_) receptors (InsP_3_Rs) and transient receptor potential vanilloid 4 (TRPV4) channels, which are, respectively, located on the endoplasmic reticulum (ER) and the plasma membrane (PM) [4,7,8]. The ensuing Ca^2+^-dependent recruitment of endothelial nitric oxide (NO) synthase (eNOS) is instrumental to generating the gasotransmitter NO [9], which dilates upstream arterioles and redirects local CBF through the active capillary branches [4]. NO is also fundamental in maintaining basal CBF [10] and initiating NVC upon somatosensory stimulation in the human brain [11]. Human brain cECs can release NO in response to a variety of neurotransmitters and neuromodulators [12,13,14,15,16], which evoke an increase in intracellular Ca^2+^ concentration to engage the eNOS. The Ca^2+^ response to neurohumoral mediators is triggered by ER Ca^2+^ mobilization through InsP_3_Rs and lysosomal Ca^2+^ release through two-pore channels (TPCs) and is sustained over time by store-operated Ca^2+^ entry (SOCE) [12,13,14,15,16]. Conversely, ryanodine receptors are absent in hCMEC/D3 cells and do not contribute to agonist-induced Ca^2+^ signals [12]. Targeting the endothelial Ca^2+^-handling machinery could, therefore, provide an alternative approach to rescue NO signalling, CBF and NVC in cerebrovascular disorders [17,18].

Allyl isothiocyanate (AITC) is an organosulfur phytochemical compound that is an abundant constituent of common cruciferous vegetables, such as broccoli, cabbage, mustard, brussels sprouts and cauliflower [19,20]. AITC induced vasodilation in rat cerebral arteries in both ex vivo brain slices [20,21] and the brain of living animals [22] via an endothelial-dependent mechanism. The hemodynamic response to AITC in rat cortical vasculature requires extracellular Ca^2+^ entry via Transient Receptor Potential Ankyrin 1 (TRPA1) channels [20]. Nevertheless, AITC has also been found to induce intracellular Ca^2+^ signals via a TRPA1-independent mechanism, which involves the production of reactive oxygen species (ROS). However, the sources(s) of AITC-induced ROS-dependent Ca^2+^ signals remain(s) to be identified [23,24,25]. Similarly, it is still unknown whether ROS may support the endothelial Ca^2+^ response to AITC and whether this Ca^2+^ signal leads to NO release in human brain cECs.

In the present investigation, we sought to assess whether and how AITC evokes Ca^2+^-dependent NO release in hCMEC/D3 cells, which represent the most widespread model of human cerebrovascular cECs [26,27,28,29]. We found that AITC induced NO production through the ROS-dependent inhibition of plasma membrane Ca^2+^-ATPase (PMCA) activity, which resulted in the accumulation of intracellular Ca^2+^ and eNOS engagement. These findings support the view that AITC administration represents a promising strategy to rescue NO signalling in cerebral disorders associated to endothelial dysfunction.

## 2. Materials and Methods

### 2.1. Cell Culture

Human cerebral microvascular endothelial cells (hCMEC/D3) were obtained from Institut National de la Santé et de la Recherche Médicale (INSERM, Paris, France). hCMEC/D3 cells cultured between passage 25 and 35 were used. As described in [30], the cells were seeded at a concentration of 27.000 cells/cm^2^ and grown in tissue culture flasks coated with 0.1 mg/mL rat tail collagen type 1, in the following medium: EBM-2 medium (Lonza, Basel, Switzerland) supplemented with 5% fetal bovine serum, 1% Penicillin-Streptomycin, 1.4 μM hydrocortisone, 5 μg/mL ascorbic acid, 1/100 chemically defined lipid concentrate (Life Technologies, Milan, Italy), 10 mM HEPES and 1 ng/mL basic fibroblast growth factor. The cells were cultured at 37 °C, 5% CO_2_-saturated humidity.

### 2.2. Solutions

Physiological salt solution (PSS) had the following composition (in mM): 150 NaCl, 6 KCl, 1.5 CaCl_2_, 1 MgCl_2_, 10 glucose, 10 HEPES. In Ca^2+^-free solution (0Ca^2+^), Ca^2+^ was substituted with 2 mM NaCl, and 0.5 mM EGTA was added. Solutions were titrated to pH 7.4 with NaOH. The osmolality of PSS as measured with an osmometer (Wescor 5500, Logan, UT, USA) was 300–310 mOsm/L.

### 2.3. [Ca^2+^]_i_, NO and ROS Imaging

AITC-TRPA1-mediated changes in [Ca^2+^]_i_ were monitored in hCMEC/D3 cells loaded with the selective Ca^2+^-fluorophore, loaded with 4 µM fura-2 acetoxymethyl ester (Fura-2/AM; 1 mM stock in dimethyl sulfoxide) in PSS for 30 min at 37 °C and 5% CO_2_, as described in [12]. After washing in PSS, the coverslip was fixed to the bottom of a Petri dish and the cells were observed by an upright epifluorescence Axiolab microscope (Carl Zeiss, Oberkochen, Germany), usually equipped with a Zeiss ×40 Achroplan objective (water-immersion, 2.0 mm working distance, 0.9 numerical aperture). The cells were excited alternately at 340 and 380 nm, and the emitted light was detected at 510 nm. A first neutral density filter (1 or 0.3 optical density) reduced the overall intensity of the excitation light, and a second neutral density filter (optical density = 0.3) was coupled to the 380 nm filter to approach the intensity of the 340 nm light. A round diaphragm was used to increase the contrast. The excitation filters were mounted on a filter wheel (Lambda 10, Sutter Instrument, Novato, CA, USA). Custom software, working in the LINUX environment, was used to drive the camera (Extended-ISIS Camera, Photonic Science, Millham, UK) and the filter wheel, and to measure and plot online the fluorescence from 10 up to 40 rectangular “regions of interest” (ROIs). Each ROI was identified by a number. Since cell borders were not clearly identifiable, a ROI may not include the whole cell or may include part of an adjacent cell. Adjacent ROIs never superimposed. [Ca^2+^]_i_ was monitored by measuring, for each ROI, the ratio of the mean fluorescence emitted at 510 nm when exciting alternatively at 340 and 380 nm (F_340_/F_380_). An increase in [Ca^2+^]_i_ causes an increase in the ratio [31]. Ratio measurements were performed and plotted online every 3 s. The experiments were performed at room temperature (22 °C).

To evaluate NO release, hCMEC/D3 cells were loaded with 4-Amino-5-methylamino-2′,7′-difluorofluorescein diacetate (DAF-FM) (1 µM) for 60 min in PSS at 22 °C, as illustrated in [30]. DAF-FM fluorescence was measured by using the same imaging setup described above for Ca^2+^ measurements but with a different filter set, i.e., excitation at 480 nm and emission at 535 nm wavelength (emission intensity denoted as NO_i_ (F_535_/F_0_)). The changes in DAF-FM fluorescence evoked by extracellular stimulation were recorded and plotted online every 5 s. Measurements of NO were performed at 22 °C.

To measure ROS production, hCMEC/D3 cells were loaded with 2′,7′-dichlorodihydrofluorescein diacetate (H_2_DCF–DA) (2 µM) in PSS for 30 min at 22 °C, as recently described [32,33]. After washout of the excess fluorophore from the extracellular solution, H2DCF-DA fluorescence of the probes was recorded (excitation/emission wavelengths, 490/520 nm) by using the same imaging setup described above (denoted as ROS (F_491_/F_0_)). ROS production was measured at 22 °C.

The cellular production of NO and ROS was reported as relative fluorescence (F/F_0_) of DAF (for NO) and H_2_DCF-Da (for ROS), where F is the fluorescence intensity obtained during recordings and F_0_ is the basal fluorescence intensity.

### 2.4. Electrophysiological Recordings

The presence of AITC-evoked membrane current was assessed by using a Port-a-patch planar patch-clamp system (Nanion Technologies, Munich, Germany) in the whole-cell, voltage-clamp or current-clamp configurations, at room temperature (22 °C), as described in [16]. Cultured cells (2–3 days after plating) were detached with Detachin and suspended at a cell density of 1–5 × 10^6^ cells/mL in external recording solution containing (in mM): 145 NaCl, 2.8 KCl, 2 MgCl_2_, 10 CaCl_2_, 10 HEPES and 10 D-glucose (pH = 7.4). Suspended cells were placed on the NPC© chip surface, and the whole-cell configuration was achieved. Internal recording solution, containing (in mM) 10 CsCl, 110 CsF, 10 NaCl, 10 HEPES and 10 EGTA (pH = 7.2, adjusted with CsOH), was deposited in recording chips, having resistances of 3–5 MΩ. The bioelectrical response to agonist (AITC or NS309) stimulation was recorded in the current-clamp mode or in the voltage-clamp mode by using an EPC-10 patch-clamp amplifier (HEKA, Munich, Germany). Changes in the membrane potential were measured in the current-clamp mode, as shown in [34], whereas the current-to-voltage relationship of agonist-evoked membrane current was evaluated in the voltage-clamp mode by applying, every 1 s, voltage ramps ranging from −100 mV to +100 mV from a holding potential of −70 mV, as described in [35,36]. Immediately after the whole-cell configuration was established, the cell capacitance and the series resistances (<10 MΩ) were measured. During recordings, these two parameters were measured, and if exceeding ≥10% with respect to the initial value, the experiment was discontinued [37]. Liquid junction potential and capacitive currents were cancelled using the automatic compensation of the EPC-10. Data were filtered at 10 kHz and sampled at 5 kHz [38].

### 2.5. Immunoblotting

Total proteins from primary hCMEC/D3 cells were treated with a RIPA buffer (150 mM NaCl, 0.5% sodium deoxycholate, 0.1% SDS, 0.1% Triton X-100 and 50 mM Tris-HCl, pH 8) supplemented with the protease inhibitor cocktail cOmplete (cOmplete Tablets EASYpack, 04693116001; Merck, Milan, Italy). Total proteins were solubilized in a Laemmli buffer, and 20 µg of proteins was separated by SDS-PAGE using precast gel electrophoresis (4–20% Mini-PROTEAN TGX Stain-Free Gels, Bio-Rad, Segrate, Italy) [12,39] and blotted onto the PVDF Membrane (Trans-Blot Turbo Transfer Pack, #1704156, Bio-Rad, Segrate, Italy) with the Trans-Blot Turbo Transfer System (#1704150, Bio-Rad, Segrate, Italy). After blocking for 1 h at RT in Tris-buffered saline containing 5% nonfat dry milk and 0.1% Tween (blocking solution), membranes were incubated overnight with anti-TRPA1 rabbit antibody (PA146159, 1:500 dilution; Thermo Fisher Scientific, Monza, Italy) in a blocking solution. After washing, the membranes were incubated for 1 h with goat anti-rabbit IgG antibody, peroxidase-conjugated (1:100,000; AP132P; Millipore part of Merck S.p.a., Vimodrone, Italy). Chemiluminescence detection of the bands was performed by incubating the blots with the Westar Supernova Western blotting detection system (CYANAGEN, Bologna, Italy), and the molecular weights were identified using pre-stained molecular weight markers (#161-0376, Bio-Rad Laboratories, Hercules, CA, USA). Anti-β2-microglobulin (B2M) rabbit monoclonal (ab75853, 1:10,000; Abcam, Cambridge, UK) was used on the stripped membrane [12]. Protein bands were detected with the iBright™ CL1000 Imaging System (Thermo Fisher Scientific, Monza, Italy).

### 2.6. Statistics

All the data have been obtained from hCMEC/D3 cells from three independent experiments. The amplitude of agonist-evoked Ca^2+^ signals was measured as the difference between the ratio at the Ca^2+^ peak and the mean ratio of 30 s baseline before the peak. The dose–response data in Figure 2B were fitted by using the following equation:(1)$$Y=\frac{100}{1+\frac{{EC}_{50}}{\left[AITC\right]}}$$
where *Y* is the response (relative to the amplitude of the Ca^2+^ response), [AITC] is the AITC concentration and *EC*_50_ is the half-maximal effective concentration. The decay time of cyclopiazonic acid (CPA)-evoked Ca^2+^ signals was computed as the time in which [Ca^2+^]_i_ decayed from 80% to 20% of its peak amplitude (t_80-20_) [40,41]. The amplitude of AITC-induced NO production was evaluated as the difference between the maximal increase in DAF-FM fluorescence and the average of 1 min baseline before the peak. The amplitude of AITC-induced ROS production was evaluated as the difference between the maximal increase in H_2_DCF-DA fluorescence and the average of 1 min baseline before the peak. Statistical significance (*p* < 0.05) was evaluated by two-tailed Student’s *t*-test for unpaired observations or one-way ANOVA analysis followed by the post hoc Dunnett’s test, as appropriate. Data relative to Ca^2+^, NO and ROS signals are presented as mean ± SE, whereas the number of cells analyzed is indicated in the corresponding bar histograms. The tracings shown in each figure are representative of the Ca^2+^, NO and ROS signals evoked by each agonist in three independent experiments for each condition.

### 2.7. Chemicals

Fura-2/AM and DAF-FM/AM were purchased from Molecular Probes (Molecular Probes Europe BV, Leiden, The Netherlands). BTP-2 was obtained from Tocris (Bristol, UK). All other chemicals were of analytical grade and obtained from Sigma Chemical Co. (St. Louis, MO, USA).

## 3. Results

### 3.1. AITC Evokes Ca^2+^-Dependent NO Release in hCMEC/D3 Cells

AITC (30 µM) evoked a slowly rising increase in DAF-FM fluorescence (Figure 1A), which reflected NO release and developed after a latency of 299.3 ± 10.8 s (*n* = 134). AITC-evoked NO release was significantly (*p* < 0.05) reduced by L-NIO di-hydrochloride (L-NIO; 50 µM), a selective blocker of eNOS activity [15], (Figure 1A,B). Furthermore, AITC-evoked NO release was prevented by buffering intracellular Ca^2+^ with BAPTA-AM (20 µM) (Figure 1A,B). Overall, these findings indicate that AITC stimulates NO production in a Ca^2+^-dependent manner in hCMEC/D3 cells.

### 3.2. AITC Evokes TRPA1-Independent Ca^2+^ Signals in hCMEC/D3 Cells

In order to assess whether AITC evokes an increase in [Ca^2+^]_i_, hCMEC/D3 cells were loaded with the Ca^2+^-sensitive fluorophore Fura-2/AM. AITC induced a slow increase in [Ca^2+^]_i_ that started at 3 µM (Figure 2A,B), reached the maximal response at between 300 µM and 1 mM (Figure 2A,B) and displayed an EC_50_ of 13.77 µM (Figure 2B). The latency of the Ca^2+^ response to 30 µM AITC was 125.2 ± 5.2 s (*n* = 105), which was significantly (*p* < 0.0001, Student’s *t*-test) faster than the latency of the accompanying NO signal (see above). Extensive washout of the agonist from the perfusate did not promote the recovery of the [Ca^2+^]_i_ to the baseline at each concentration probed (Figure 2A). This observation suggests that AITC promotes an irreversible modification either of the target channel or of the Ca^2+^-clearing mechanisms. AITC induces an increase in [Ca^2+^]_i_ by activating TRPA1 channels on the plasma membrane [20,21] or via cytosolic ROS production [23,24,25]. Immunoblotting identified a major band of ~140 kDa, which is the predicted molecular weight of TRPA1 protein [42], in hCMEC/D3 cells (Figure 2C). However, the Ca^2+^ response to AITC (30 µM) was not affected by HC-030031 (30 µM) (Figure 2D), which is regarded to be among the most selective TRPA1 channel inhibitors [20,42,43]. To further investigate whether the TRPA1 channel was sensitive to AITC in hCMEC/D3 cells, we exploited the planar patch-clamp technique [16]. AITC-dependent Ca^2+^ entry has been shown to hyperpolarize mouse brain microvascular endothelial cells by recruiting the small-and intermediated conductance Ca^2+^-activated K^+^ channels (respectively, SK_Ca_ and IK_Ca_) [20,44]. However, AITC (30 µM) failed either to activate transmembrane currents (Appendix A) or to induce membrane hyperpolarization (Appendix A) in hCMEC/D3 cells. Nevertheless, NS309 (10 µM), a selective SK_Ca_/IK_Ca_ opener [45,46], activated an outwardly rectifying current reverting at ~−70 mV (Appendix A), which is the predicted equilibrium potential for K^+^ under our conditions. Furthermore, NS309 (10 µM) caused a rapid and reversible membrane hyperpolarization (Appendix A), which is consistent with the expression of SK_Ca_/IK_Ca_ in hCMEC/D3 cells. Therefore, the lack of effect of AITC is not due to the absence of SK_Ca_/IK_Ca_ channels in these cells. Nevertheless, 4-Hydroxy-nonenal (4-HNE), a metabolite of lipid peroxidation that may serve as an endogenous agonist of TRPA1 in mouse brain microcirculation [5,21], evoked a rapid increase in [Ca^2+^]_i_ in hCMEC/D3 cells that was inhibited by HC-030031 (30 µM) (Appendix A). Overall, these findings indicate that TRPA1 is functional but is unlikely to support AITC-evoked Ca^2+^ signals in hCMEC/D3 cells.

### 3.3. Cytosolic ROS Mediate the Ca^2+^ Response to AITC in hCMEC/D3 Cells

The role of cytosolic ROS in the Ca^2+^ response to AITC (30 µM) was evaluated in hCMEC/D3 cells pretreated with n-acetylcysteine (NAC; 1 mM, 1 h), a thiol-containing compound that serves as a direct ROS scavenger [47]. The AITC-evoked Ca^2+^ signal was significantly (*p* < 0.05) reduced in the presence of NAC, although removal of the synthetic antioxidant from the perfusate fully restored the amplitude of the increase in [Ca^2+^]_i_ (Figure 3A,B). Conversely, the Ca^2+^ response to AITC was not impaired by MitoTEMPO (10 µM) (Figure 3C,D), a selective scavenger of mitochondrial ROS [47]. In accord, AITC (30 µM) caused an immediate and long-lasting increase in cytosolic ROS production (Figure 3E), which was evaluated in hCMEC/D3 cells loaded with the ROS-sensitive fluorescent indicator H_2_DCF–DA and exhibited a latency of 20.7 *±* 1.8 s (*n* = 72). This delay was significantly (*p* < 0.0001, One-way ANOVA followed by the post hoc Dunnett’s test) faster than the latency of the accompanying Ca^2+^ and NO signals (see above). AITC-induced ROS production was abolished by NAC (1 mM) (Figure 3E,F), but not MitoTEMPO (10 µM) (Figure 3E,F). Overall, these findings demonstrate that the Ca^2+^ response to AITC is triggered by cytosolic ROS production.

### 3.4. The Ca^2+^ Response to AITC Requires Cytosolic Ca^2+^ and Is Sustained by SOCE

In hCMEC/D3 cells, intracellular Ca^2+^ signals leading to NO release are shaped by ER Ca^2+^ release and lysosomal Ca^2+^ mobilization and sustained over time by SOCE [12,13,14]. Under 0Ca^2+^ conditions, AITC (30 µM) was still able to evoke a slow increase in [Ca^2+^]_i_ (Figure 4A), which attained a significantly (*p* < 0.05) lower amplitude as compared to that measured in the presence of Ca^2+^ (Figure 4B). Thus, both cytosolic and extracellular Ca^2+^ sources are required to shape the Ca^2+^ response to AITC. Intriguingly, also in the absence of extracellular Ca^2+^, the increase in [Ca^2+^]_i_ evoked by AITC did not decline to the baseline, as otherwise observed in response to physiological neurotransmitters [12,13].

SOCE represents the main Ca^2+^-entry pathway that sustains neurotransmitters-induced NO release in hCMEC/D3 cells [12,13]. The Ca^2+^ response to AITC (30 µM) was reversibly reduced by three distinct SOCE inhibitors (Figure 4C,D): BTP-2 (20 µM), La^3+^ (10 µM) and Gd^3+^ (10 µM) [48]. Thus, SOCE also sustains AITC-evoked Ca^2+^ signals in these cells. Note that removal of the SOCE inhibitors from the perfusate caused an increase in the amplitude of the increase in [Ca^2+^]_i_ (Figure 4C).

We then focused our attention on the intracellular Ca^2+^ pool targeted by AITC (30 µM). AITC-evoked intracellular Ca^2+^ mobilization was not attenuated by depleting either the ER Ca^2+^ reservoir with CPA (10 µM) or the lysosomal Ca^2+^ store with nigericin (50 µM) (Figure 4E,F). CPA depletes the ER Ca^2+^ store by inhibiting SERCA activity and preventing cytosolic Ca^2+^ sequestration into ER lumen [48], whereas nigericin is a protonophore that prevents cytosolic Ca^2+^ accumulation into lysosomal lumen by inhibiting the H^+^-dependent Ca^2+^ uptake mechanism [49,50]. Additionally, AITC-evoked Ca^2+^ mobilization was not affected by blocking inositol-1,4,5-trisphosphate receptors (InsP_3_Rs), which represent the sole ER Ca^2+^-releasing channel in hCMEC/D3 cells [12], with 2-Aminoethyl Diphenylborinate (2-APB; 50 µM) (Figure 4E,F). Likewise, AITC-evoked Ca^2+^ release was insensitive to depletion of the mitochondrial Ca^2+^ pool with the protonophore FCCP (10 µM) (Figure 4E,F). Thus, AITC is unlikely to increase the [Ca^2+^]_i_ by targeting the main cytosolic Ca^2+^ reservoirs involved in endothelial Ca^2+^ signalling. It turns out that AITC-evoked SOCE activation is not driven by ER Ca^2+^ depletion.

### 3.5. AITC Inhibits PMCA Activity to Increase the [Ca^2+^]_i_ in hCMEC/D3 Cells

The observation that the Ca^2+^ response to AITC does not decline to the baseline either upon removal of the agonist or during prolonged stimulation under 0Ca^2+^ conditions suggests that AITC inhibits the Ca^2+^-clearing mechanisms that maintain the resting [Ca^2+^]_i_ in hCMEC/D3 cells. These include sarco-endoplasmic Ca^2+^-ATPase (SERCA) in the ER, the plasma membrane Ca^2+^-ATPase (PMCA) and the Na^+^/Ca^2+^ exchanger (NCX) on the plasma membrane [18]. AITC-induced inhibition of the Ca^2+^-clearing mechanisms would result in the accumulation of cytosolic Ca^2+^ and thereby in an increase in [Ca^2+^]_i_. Nevertheless, SERCA inhibition does not account for the Ca^2+^ response to AITC since this fully occurs upon inhibition of SERCA activity with CPA (see Figure 4E,F).

To assess whether AITC targets NCX or PMCA or both, we exploited the strategy described to evaluate the H_2_O_2_-dependent blockade of PMCA activity in mouse parotid acinar cells [41]. CPA evokes a robust increase in [Ca^2+^]_i_ by inhibiting SERCA and thereby inducing passive ER Ca^2+^ efflux in the cytoplasm [12]. Under 0Ca^2+^ conditions, the Ca^2+^ response to CPA (10 µM) declines to the baseline because of cytosolic Ca^2+^ removal by NCX, PMCA or mitochondrial Ca^2+^ uniporter (MCU) (Figure 5A). Therefore, we first measured the rate of decline of the CPA-evoked increase in [Ca^2+^]_i_ in the absence and presence of SEA0400 (10 µM), vanadate (VO3^−^; 500 µM) and Ru360 (10 µM), which, respectively, block NCX [51], PMCA [51] and MCU [52] (Figure 5A). Preliminary recordings showed that the green fluorescence of carboxyeosin, which also inhibits endothelial PMCA activity [53], was too bright and interfered with Fura-2 fluorescence in hCMEC/D3 cells. The decay rate of CPA-evoked intracellular Ca^2+^ release was slightly, although significantly (*p* < 0.05), slowed by either SEA0400 or Ru360 (Figure 5A,B). Conversely, in the presence of VO3^−^, the increase in [Ca^2+^]_i_ evoked by CPA did not even return to the baseline and resulted in a long-lasting plateau (Figure 5A). The value of t_80-20_, therefore, could not be measured under these conditions (Figure 5B). Once assessed that PMCA is the main factor responsible for the decay of the CPA-evoked intracellular Ca^2+^ release, we evaluated the effect of AITC (30 µM). AITC mimicked the effect of VO3^−^ by converting the transient increase in [Ca^2+^]_i_ evoked by CPA under 0Ca^2+^ conditions into a biphasic elevation in [Ca^2+^]_i_ that did not enable us to measure the τ_80-20_ (Figure 5B). The amplitude of the long-lasting plateau caused by VO3^−^ and AITC in hCMEC/D3 cells stimulated with CPA is displayed in Figure 5C. Furthermore, pretreatment with AITC significantly (*p* < 0.05) reduced the amplitude of the Ca^2+^ response to CPA (Figure 5D). These findings suggest that AITC increases [Ca^2+^]_i_ by inhibiting PMCA activity. Consistently, VO3^−^ (500 µM) caused a slow increase in [Ca^2+^]_i_ (Figure 5E) and prevented the intracellular Ca^2+^ response to AITC (Figure 5E). In contrast, neither SEA0400 (10 µM) nor Ru360 (10 µM) increased the resting [Ca^2+^]_i_ (Figure 5E) or impaired AITC-induced Ca^2+^ signals under 0Ca^2+^ conditions (Figure 5E). The statistical analysis of these data is displayed in Figure 5F. hCMEC/D3 cells were then challenged with hydrogen peroxide (H_2_O_2_; 100 µM) under 0Ca^2+^ conditions. Exogenous administration of H_2_O_2_ caused a slow increase in [Ca^2+^]_i_ that mimicked and prevented the subsequent Ca^2+^ response to AITC (Figure 5G,H). In addition, AITC failed to slow down and convert into a long-lasting plateau the decay phase of CPA-evoked Ca^2+^ transient in hCMEC/D3 cells pretreated with NAC (1 mM) (Appendix A). Finally, exposure of hCMEC/D3 cells with H_2_O_2_ converted the transient increase in [Ca^2+^]_i_ induced by CPA (10 µM) into a biphasic Ca^2+^ signal that presented a long-lasting plateau (Appendix A), as observed with VO3- and AITC.

Overall, these findings confirm that the Ca^2+^ response evoked by AITC in the absence of extracellular Ca^2+^ is due to the ROS-dependent inhibition of PMCA activity. This in turn interferes with the extrusion of cytosolic Ca^2+^ across the plasma membrane and results in a progressive elevation in [Ca^2+^]_i_. PMCA can be tightly coupled to store-operated channels and is the main responsible for clearing the incoming Ca^2+^ across the plasma membrane [54,55]. SOCE is constitutively activated in hCMEC/D3 cells [12]. Therefore, ROS-dependent PMCA inhibition could also unmask this background Ca^2+^ entry route and thereby explain the blocking effects of BTP-2 and trivalent cations on AITC-evoked Ca^2+^ signals.

### 3.6. AITC-Evoked NO Release Requires ROS-Dependent Inhibition of PMCA Activity and SOCE

In agreement with the model described above, AITC (30 µM) failed to cause a detectable NO signal in hCMEC/D3 cells pretreated with NAC (1 mM) (Figure 6A). Furthermore, AITC-evoked NO release was significantly (*p* < 0.05) reduced in the presence of either VO3^−^ (500 µM) (Figure 6A) or BTP-2 (20 µM) (Figure 6A), which, respectively, inhibit PMCA and SOCE. The statistical analysis of these findings is reported in Figure 6B. Notably, the acute addition of VO3^−^ caused a slow increase in DAF-FM fluorescence (Figure 6C), which was reminiscent of that caused by AITC (30 µM) and was inhibited by L-NIO (50 µM) (Figure 6C,D) and BAPTA (20 µM) (Figure 6C,D). Finally, exogenous administration of H_2_O_2_ (100 µM) was also able to induce NO release in hCMEC/D3 cells (Figure 6E) and, as reported for AITC and VO3^−^, H_2_O_2_-induced NO production was inhibited by L-NIO (50 µM) (Figure 6C,D) and BAPTA (20 µM). These findings confirm that AITC evokes NO release in hCMEC/D3 cells by causing cytosolic ROS production, which inhibits PMCA to increase the [Ca^2+^]_i_ and engage eNOS.

## 4. Discussion

Herein, we demonstrated that AITC, a major active constituent of cruciferous vegetables, elicits NO release in the human cerebrovascular endothelial cell line hCMEC/D3 by generating intracellular ROS that inhibit PMCA activity and thereby cause a slow increase in [Ca^2+^]_i_. This is the first demonstration of the signalling pathway whereby AITC-dependent ROS production stimulates intracellular Ca^2+^ activity in a TRPA1-independent manner [42]. We further showed that the AITC-induced increase in [Ca^2+^]_i_ leads to NO release, thereby providing the proof-of-concept that AITC administration could represent a promising strategy to prevent deficits in CBF.

It has long been known that AITC elicits vasodilation in rat cerebral arteries in ex vivo brain slices [20,21] and increases meningeal blood flow in the brain of living animals [22]. These studies showed that the hemodynamic response to AITC in rat brain vasculature was accomplished by TRPA1-mediated Ca^2+^ entry in cECs, which triggered a regenerative Ca^2+^ wave propagating back to upstream arterioles and recruiting endothelial Ca^2+^-dependent K^+^ channels to hyperpolarize adjacent vascular smooth muscle cells [20,21]. AITC-induced vasodilation in rat brain vasculature was insensitive to eNOS inhibition [20]. Nevertheless, the ion signalling machinery that regulates CBF can slightly differ between human and mice cECs [18]. Herein, we found that AITC causes a slow production of NO in hCMEC/D3 cells loaded with DAF-FM, a selective NO-sensitive fluorescent probe that is widely employed to measure NO release in both cultured cECs [12,13] and in brain microvessels [56]. It has previously been shown that AITC stimulates NO production in adult mouse ventricular cardiomyocytes [57], while this is the first demonstration that AITC evokes NO release in endothelial cells. The AITC-induced increase in DAF-FM fluorescence was suppressed by selectively inhibiting eNOS activity with L-NIO and by buffering intracellular Ca^2+^ levels with BAPTA. Therefore, we hypothesized that AITC was also able to induce NO release through an increase in the [Ca^2+^]_i_ in hCMEC/D3 cells.

Consistently, we showed that AITC caused a dose-dependent increase in [Ca^2+^]_i_ that was insensitive to agonist removal from the perfusate. Intriguingly, the EC_50_ for AITC-evoked Ca^2+^ signals (13.77 µM) was similar to those reported for AITC-evoked Ca^2+^ sparklet in rat brain cECs (4.4 µM) [21] and for AITC-evoked vasodilation in rat cerebral arteries (16.4 µM) [44]. Nevertheless, while AITC-evoked Ca^2+^ signals [21] and vasodilation [44] in rat brain vasculature were sensitive to TRPA1 inhibition, HC-030031 did not affect the Ca^2+^ response to AITC in hCMEC/D3 cells. Intriguingly, TRPA1 protein was expressed in these cells, thereby suggesting that AITC-dependent TRPA1 activation in hCMEC/D3 cells is strongly inhibited by intracellular modulators, such as phosphatidylinositol-4,5-bisphosphate and inorganic polyphosphates [42]. In addition, AITC was still able to evoke robust, albeit slightly lower, Ca^2+^ signals under 0Ca^2+^ conditions in hCMEC/D3 cells, while AITC-evoked Ca^2+^ sparklets were suppressed by removal of extracellular Ca^2+^ and TRPA1 inhibition with HC-030031 in rat brain cECs [21]. Therefore, AITC does not increase the [Ca^2+^]_i_ by activating TRPA1 in hCMEC/D3 cells.

Early studies showed that AITC can induce intracellular Ca^2+^ signals that are not supported by TRPA1-mediated Ca^2+^ entry but require cytosolic ROS production in several cancer cell lines [23,24]. Herein, we confirmed that AITC induced rapid cytosolic, but not mitochondrial, ROS production and failed to induce a full increase in [Ca^2+^]_i_ in hCMEC/D3 cells pretreated with the widely employed antioxidant NAC. Cytosolic ROS could mobilize the intracellular Ca^2+^ pools that are located in both the ER and the acidic lysosomal compartment by directly gating ER-located InsP_3_Rs or the lysosomal TRP Mucolipin 1 channel [58,59]. However, AITC still increased the [Ca^2+^]_i_ in hCMEC/D3 cells bathed in the absence of extracellular Ca^2+^ and pretreated with either CPA or nigericin that, respectively, deplete the ER and lysosomal Ca^2+^ stores. Mitochondria represent an additional Ca^2+^ reservoir, although it is unlikely to support endothelial Ca^2+^ signalling by mobilizing matrix Ca^2+^ [60]. In accord, depleting mitochondrial Ca^2+^ with FCCP did not affect AITC-evoked intracellular Ca^2+^ mobilization in hCMEC/D3 cells. These unexpected findings led us to hypothesize that AITC did not target an endogenous Ca^2+^ reservoir to increase the [Ca^2+^]_i_ under 0Ca^2+^ conditions, but rather inhibited the Ca^2+^-clearing machinery. This hypothesis was also supported by the evidence that the Ca^2+^ influx component of AITC-evoked Ca^2+^ signals was hampered by the selective blockade of SOCE with La^3+^, Gd^3+^ and BTP-2. These compounds inhibit Orai1 [48], which represents the pore-forming subunits of Ca^2+^-selective store-operated channels [48] and mediates SOCE in hCMEC/D3 cells [12]. However, ROS are not able to directly gate extracellular Ca^2+^ entry through Orai1 [58]. Intriguingly, Orai1 channels are closely coupled to PMCA in vascular endothelial cells such that incoming Ca^2+^ is rapidly extruded across the plasma membrane by PMCA-mediated Ca^2+^ removal [48,55]. SOCE is tonically activated in hCMEC/D3 cells [12]. Therefore, we speculated that AITC-evoked ROS production could inhibit PMCA activity, thereby progressively leading to higher [Ca^2+^]_i_ and unmasking constitutive SOCE.

Early studies showed that ROS do inhibit PMCA activity and cause a slow and irreversibly sustained increase in [Ca^2+^]_i_ similar to that elicited by AITC in hCMEC/D3 cells [41,61]. Therefore, we first evaluated whether AITC slowed the decay rate of CPA-evoked intracellular Ca^2+^ release, which is largely due to PMCA activation in hCMEC/D3 cells, as shown by our preliminary characterization (Figure 5A). Pretreatment with AITC mimicked the effect of VO3^−^ by preventing the [Ca^2+^]_i_ from returning to basal levels after the initial rise in [Ca^2+^]_i_ induced by CPA in the absence of extracellular Ca^2+^. Furthermore, tempering AITC-induced ROS production with NAC did not remarkably affect the recovery of CPA-evoked intracellular Ca^2+^ release. Moreover, acute addition of VO3^−^ was, per se, able to induce a sustained increase in [Ca^2+^]_i_ in hCMEC/D3 cells, while blocking NCX with SEA0400 or preventing mitochondrial Ca^2+^ entry with Ru360 did not alter the basal [Ca^2+^]_i_ and did not affect the AITC-evoked Ca^2+^ response. Finally, exogenous H_2_O_2_ caused a slow Ca^2+^ signal that resembled that induced by either AITC or VO3^−^, converted the transient increase in [Ca^2+^]_i_ induced by CPA into a long-lasting biphasic signal and prevented the subsequent Ca^2+^ response to AITC. These findings are consistent with a model according to which PMCA is primarily involved in maintaining resting [Ca^2+^]_i_, such that its inhibition by AITC-induced cytosolic ROS production or VO3^−^ results in cytosolic Ca^2+^ accumulation. In accord, the pharmacological [62] or genetic [63] blockade of PMCA activity may increase [Ca^2+^]_i_ in vascular endothelial cells. Overall, our data indicate that, in hCMEC/D3 cells, AITC inhibits PMCA activity through the production of cytosolic ROS, thereby attenuating Ca^2+^ extrusion across the plasma membrane and resulting in a slow increase in [Ca^2+^]_i_. In addition, PMCA is predicted to remove Ca^2+^ that enters into the cytosol through basally active store-operated channels [12], such that constitutive SOCE further enhances cytosolic Ca^2+^ accumulation. This model is supported by evidence that AITC-induced NO release was inhibited by preventing the accompanying increase in [Ca^2+^]_i_ with (1) NAC, which prevents ROS accumulation; (2) VO3^−^, which inhibits PMCA activity; and (3) BTP-2, which inhibits SOCE. In further accordance with this hypothesis, ROS production precedes the increase in [Ca^2+^]_i_ that in turn comes first NO release, as testified by the latencies of the corresponding signals. Of note, early studies demonstrated that PMCA1 and PMCA4, which are both expressed in hCMEC/D3 cells [12], negatively regulate eNOS activity in vascular endothelial cells by tethering it to a low Ca^2+^ microdomain [63,64,65].

## 5. Conclusions

This investigation demonstrated that AITC stimulates NO release from hCMEC/D3 cells, a widely employed model of human brain cECs, though an increase in [Ca^2+^]_i_ that is driven by ROS-dependent inhibition of PMCA activity. Targeting the endothelial ion channel machinery might represent an alternative strategy to restore NO signalling and rescue CBF in brain disorders [18]. Preclinical studies confirmed that intracarotid infusion of AITC caused a remarkable dilation of dural arteries and increase meningeal blood flow [22]. Future studies will have to assess whether AITC administration is also able to increase NO production in the human brain. Interestingly, the production of high levels of ROS plays a crucial role in cerebral ischemia-reperfusion injury by causing endothelial dysfunction and BBB disruption [66,67,68]. However, it has long been known that low ROS signalling regulates multiple endothelial functions and could be instrumental in rescuing endothelial dysfunction in cerebrovascular and neurological disorders [58,69,70], including atherosclerosis, acute myocardial infarction, stroke and traumatic brain injury. Therefore, the findings reported in the present investigation suggest that dietary supplementation of AITC could be more efficient to prevent ischemic events in the brain by targeting the endothelial ion signalling machinery when ROS production in cerebral microcirculation is still under tight homeostatic control.

## Figures and Tables

**Figure 1 cells-12-01732-f001:**
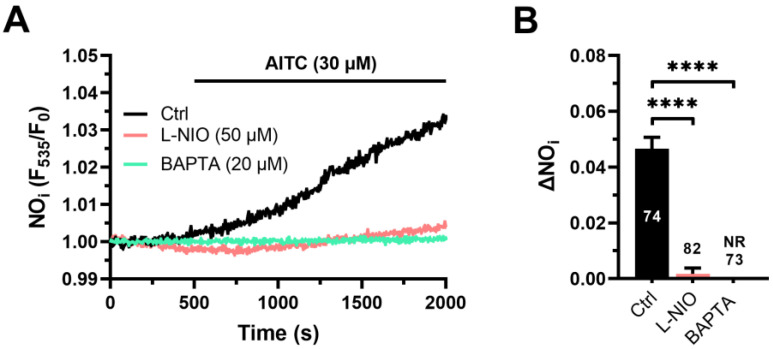
AITC-evoked Ca^2+^-dependent NO release in hCMEC/D3 cells. (**A**) AITC (30 µM) evoked a slow-rising increase in DAF-FM fluorescence, which reflected NO release and was inhibited by L-NIO (50 µM, 1 h) and BAPTA (20 µM, 2 h). In this and the following figures, AITC has been administered at the time indicated by the black bar that has been drawn above the NO, Ca^2+^, or ROS tracings. (**B**) Mean ± SE of the amplitude of AITC-evoked NO release in the absence (Ctrl) or in the presence of L-NIO or BAPTA. One-way ANOVA followed by the post hoc Dunnett’s test: **** *p* < 0.0001. Ctrl: control. NR: no response.

**Figure 2 cells-12-01732-f002:**
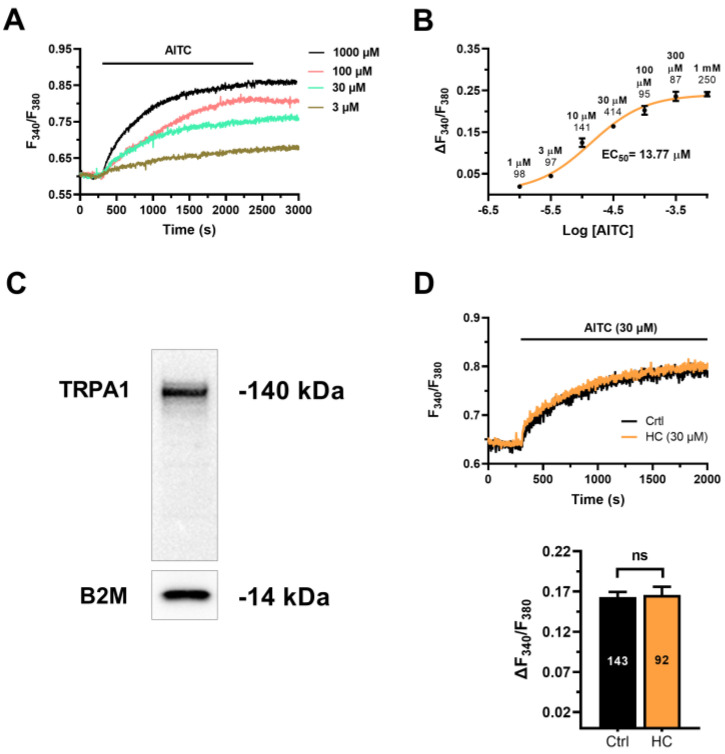
AITC elicits TRPA1-independent Ca^2+^ signals in hCMEC/D3 cells. (**A**) Intracellular Ca^2+^ signals evoked by increasing concentrations of AITC in hCMEC/D3 cells. The [Ca^2+^]_i_ never returned to the baseline upon agonist washout. The baseline of the Ca^2+^ responses has been slightly shifted to avoid tracing overlapping. (**B**) Dose-response relationship of the amplitude of AITC-evoked Ca^2+^ signals. The sigmoidal line that fits the dose -response curve was obtained by using Equation (1). (**C**) TRPA1 protein expression in hCMEC/D3 cells. Blots representative of four independent experiments are shown. Major bands of the expected molecular weights are indicated. (**D**) Upper panel, AITC-evoked Ca^2+^ signals were not inhibited by blocking TRPA1 with HC-030031 (30 µM). Lower panel, mean ± SE of the amplitude of AITC-evoked NO in the absence (Ctrl) and presence of HC-030031 (HC). NS: not significant, Student’s *t*-test.

**Figure 3 cells-12-01732-f003:**
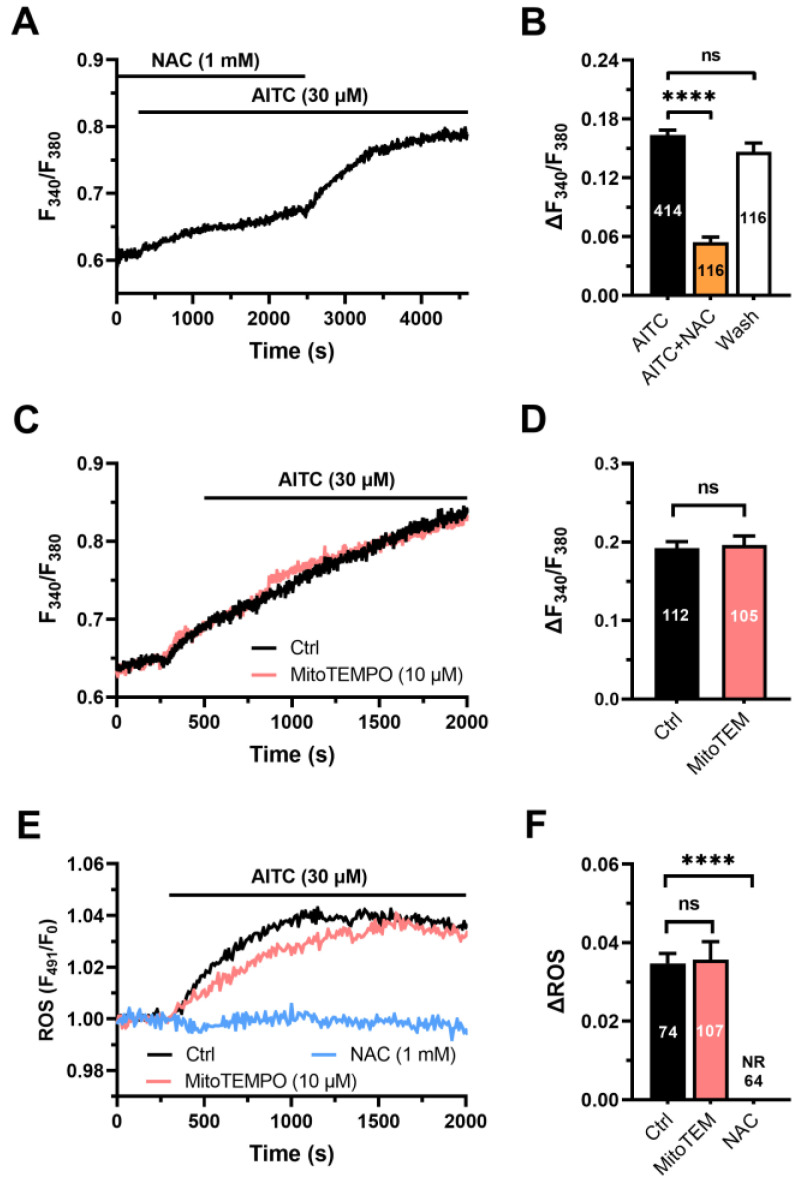
Cytosolic ROS drive AITC-evoked Ca^2+^ signals in hCMEC/D3 cells. (**A**) Intracellular Ca^2+^ signals evoked by AITC (30 µM) were dampened by the antioxidant NAC (1 mM, 1 h). NAC washout from the perfusate caused a further AITC-dependent elevation in [Ca^2+^]_i_. (**B**) Mean ± SE of the amplitude of the Ca^2+^ signals evoked by AITC in the absence (AITC) or in the presence of NAC (AITC+NAC) and upon AITC washout from the perfusate (Wash). One-way ANOVA followed by the post hoc Dunnett’s test: **** *p* < 0.0001. NS: not significant. (**C**) Intracellular Ca^2+^ signals evoked by AITC (30 µM) were not affected by scavenging mitochondrial ROS with MitoTEMPO (MitoTEM; 10 µM, 1 h). (**D**) Mean ± SE of the amplitude of AITC-evoked Ca^2+^ signals in the absence (Ctrl) or in the presence of MitoTEMPO (MitoTEM). NS: not significant, Student’s *t*-test. Ctrl: control. (**E**) AITC (30 µM) evoked robust ROS production in hCMEC/D3 cells that was sensitive to NAC (1 mM), but not to MitoTEMPO (MitoTEM; 10 µM). (**F**) Mean ± SE of the amplitude of AITC-evoked Ca^2+^ signals in the absence (Ctrl) or in the presence of MitoTEMPO (MitoTEM) or NAC. One-way ANOVA followed by the post hoc Dunnett’s test: **** *p* < 0.0001. NR: no response.

**Figure 4 cells-12-01732-f004:**
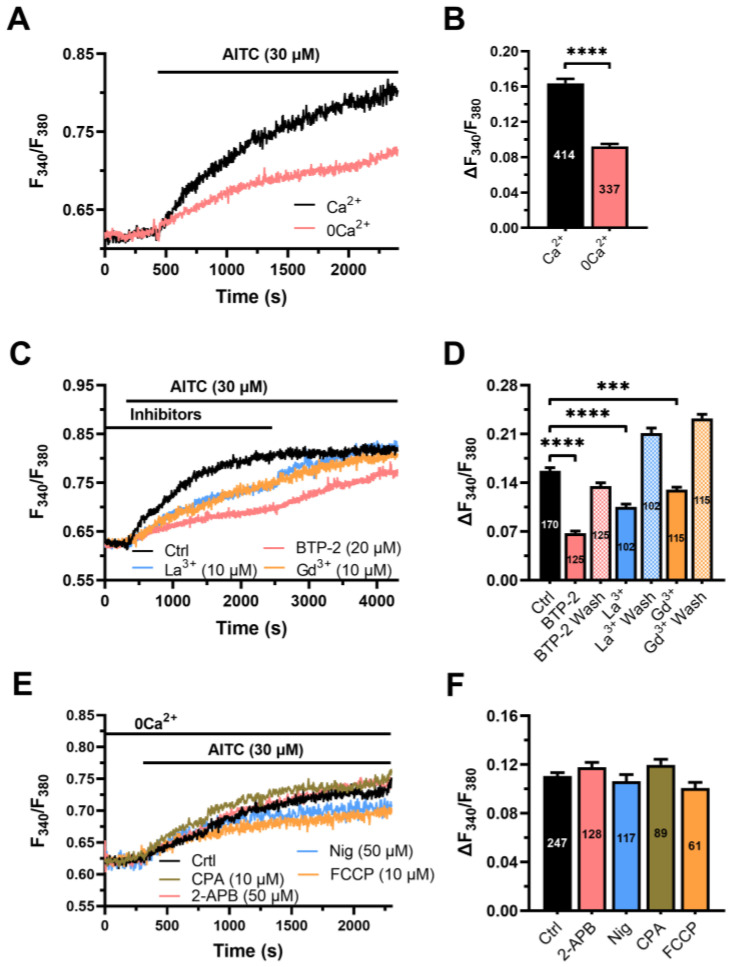
AITC-evoked Ca^2+^ signals require SOCE but not intracellular Ca^2+^ release. (**A**) Intracellular Ca^2+^ signals evoked by AITC (30 µM) were dampened in the absence of extracellular Ca^2+^ (0Ca^2+^). (**B**) Mean ± SE of the amplitude of AITC-evoked Ca^2+^ signals in the presence (Ca^2^) or in the absence of extracellular Ca^2^ (0Ca^2^). Student’s *t*-test: *** *p* < 0.001. (**C**) Intracellular Ca^2+^ signals evoked by AITC (30 µM) were dampened upon SOCE inhibition with BTP-2 (20 µM, 20 min), La^3+^ (10 µM, 20 min) and Gd^3+^ (10 µM, 20 min). Removal of each inhibitor from the perfusate caused a further AITC-dependent elevation in [Ca^2+^]_i_. (**D**) Mean ± SE of the amplitude of AITC-evoked Ca^2+^ signals under the following conditions: in the absence of any inhibitor (Ctrl); in the presence of BTP-2 (BTP-2) or after BTP-2 washout from the perfusate (Wash); in the presence of La^3+^ (La^3+^) or after La^3+^ washout from the perfusate (Wash); and in the presence of Gd^3+^ (Gd^3+^) or after Gd^3+^ washout from the perfusate (Wash). One-way ANOVA followed by the post hoc Dunnett’s test: **** *p* < 0.0001; *** *p* < 0.001. (**E**) AITC (30 µM) still evoked robust intracellular Ca^2+^ signals in the presence of CPA (10 µM, 30 min), nigericin (Nig; 50 µM, 30 min), 2-APB (50 µM, 30 min) and FCCP (10 µM, 30 min). (**F**) Mean ± SE of the amplitude of AITC-evoked Ca^2+^ signals in the absence (Ctrl) or in the presence of 2-APB, nigericin (Nig), CPA or FCCP.

**Figure 5 cells-12-01732-f005:**
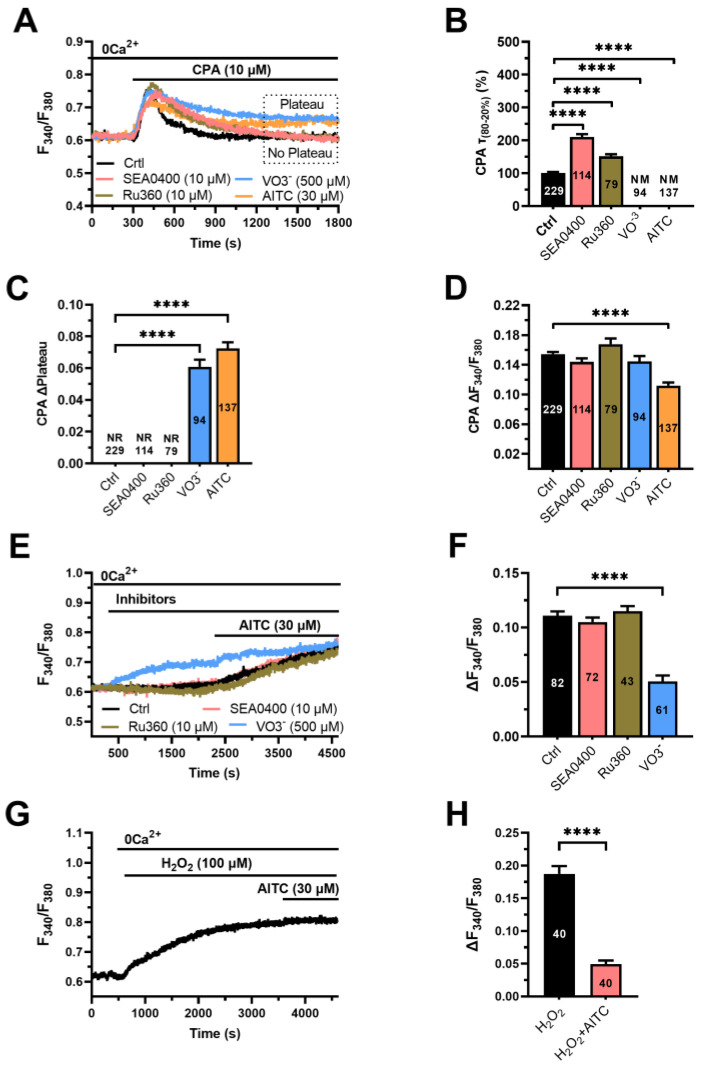
AITC-induced ROS production inhibits PMCA activity in hCMEC/D3 cells. (**A**) Intracellular Ca^2+^ release evoked by CPA (10 µM) in the absence (Ctrl) and presence of SEA0400 (10 µM, 30 min), VO3^−^ (500 µM, 30 min), Ru360 (10 µM, 30 min) and AITC (30 µM). In the presence of VO3^−^ or AITC, the [Ca^2+^]_i_ did not return to the baseline, but rather decayed to a sustained plateau level. The baselines of the Ca^2+^ tracings have been overlapped for representative purposes. (**B**) Mean ± SE of the percentage change in the value of τ_80-20_ of CPA-evoked Ca^2+^ release in the absence (Ctrl) or in the presence of SEA0400, Ru360 or VO3^−^. One-way ANOVA followed by the post hoc Dunnett’s test: **** *p* < 0.0001. NM: not measurable. (**C**) Mean ± SE of the amplitude of the long-lasting plateau evoked by CPA in the presence of VO3^−^ or AITC. One-way ANOVA followed by the post hoc Dunnett’s test: **** *p* < 0.0001. NR: no response (i.e., no plateau arising). (**D**) Mean ± SE of the amplitude of CPA-evoked intracellular Ca^2+^ release in the absence (Ctrl) or in the presence of SEA0400, Ru360 or VO3^−^. One-way ANOVA followed by the post hoc Dunnett’s test: **** *p* < 0.0001. (**E**) Exogenous administration of VO3^−^ (500 µM), but not of SEA0400 (10 µM) or Ru360 (10 µM), caused a slow increase in [Ca^2+^]_i_ that dampened the subsequent Ca^2+^ response to AITC (30 µM). (**F**) Mean ± SE of the amplitude of the Ca^2+^ signals evoked by AITC in the absence (Ctrl) or in the presence of SEA0400, Ru360 or VO3^−^. One-way ANOVA followed by the post hoc Dunnett’s test: **** *p* < 0.0001. (**G**) Exogenous administration of H_2_O_2_ (100 µM) caused a slow increase in [Ca^2+^]_i_ that dampened the subsequent Ca^2+^ response to AITC (30 µM). (**H**) Mean ± SE of the amplitude of the Ca^2+^ signals evoked by H_2_O_2_ (H_2_O_2_) or by AITC after H_2_O_2_ stimulation (AITC+ H_2_O_2_); Student’s *t*-test: **** *p* < 0.0001.

**Figure 6 cells-12-01732-f006:**
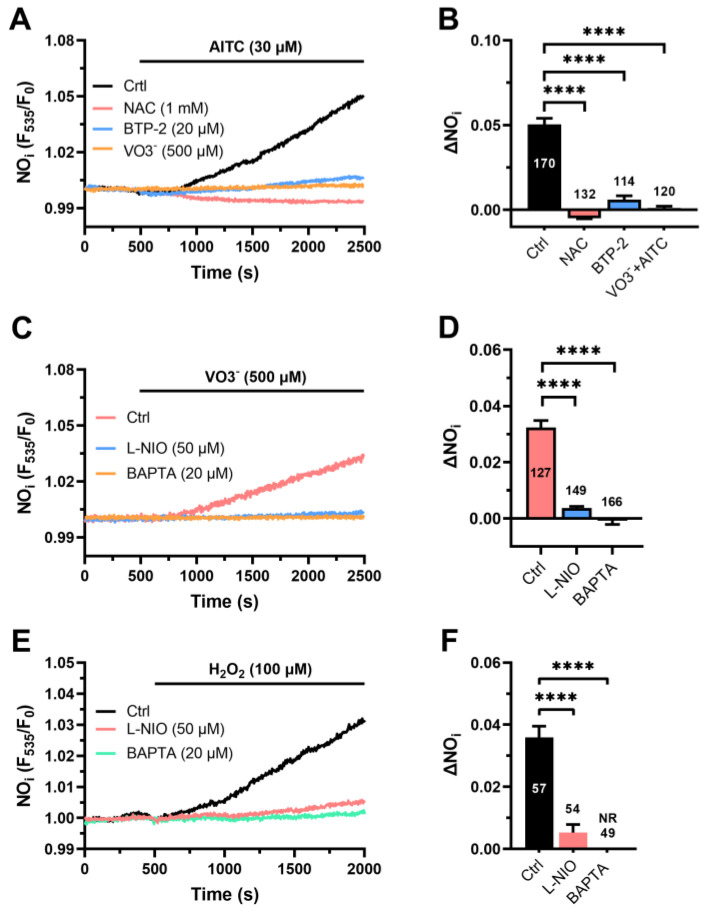
The role of AITC-evoked Ca^2+^ signals in AITC-induced NO release. (**A**) AITC (30 µM) evokes robust NO production in the absence (Ctrl), but not in the presence, of NAC (1 mM, 1 min), BTP-2 (20 µM, 20 min) and VO3^−^ (500 µM, 30 min). (**B**) Mean ± SE of the amplitude of NO release induced by AITC in the absence (Ctrl) or in the presence of NAC (NAC), BTP-2 (BTP-2) or VO3^−^. One-way ANOVA followed by the post hoc Dunnett’s test: **** *p* < 0.0001. (**C**) VO3^−^ (500 µM) evokes robust NO production in the absence (Ctrl), but not in the presence, of BAPTA (20 µM, 2 h) or L-NIO (50 µM, 1 h). (**D**) Mean ± SE of the amplitude of NO release induced by VO3^−^ in the absence (Ctrl) or in the presence of L-NIO (L-NIO) or BAPTA (BAPTA). One-way ANOVA followed by the post hoc Dunnett’s test: **** *p* < 0.0001. (**E**) Exogenous administration of H_2_O_2_ (100 µM) induces NO release in the absence (Ctrl), but not in the presence, of BAPTA (20 µM, 2 h) or L-NIO (50 µM, 1 h). (**F**) Mean ± SE of the amplitude of NO release induced by VO3^−^ in the absence (Ctrl) or in the presence of L-NIO (L-NIO) or BAPTA (BAPTA). One-way ANOVA followed by the post hoc Dunnett’s test: **** *p* < 0.0001.

## Data Availability

Original data are available upon reasonable request to the corresponding author.

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
