# Peer review of "Allyl Isothiocianate Induces Ca2+ Signals and Nitric Oxide Release by Inducing Reactive Oxygen Species Production in the Human Cerebrovascular Endothelial Cell Line hCMEC/D3"

_cells, 2023, doi:10.3390/cells12131732_

Round 1

Reviewer 1 Report

In the present investigation, Berra-Romani and coworkers examined whether and how allyl isothyocianate (AITC), a major active constituent of cruciferous vegetables, was able to induce NO release in the human cerebrovascular endothelial cell line, hCMEC/D3, which is one of the most widely used cellular models of human blood-brain barrier. AITC has long been known to induce microvessel dilation in mouse brain by activating the endothelial Ca2+-permeable channel, TRPA1. Although TRPA1 protein is present in hCMEC/D3 cells, the authors found that AITC evoked an increase in [Ca2+]i that was independent of TRPA1. Conversely, they found that the Ca2+ response to AITC was driven by early cytosolic ROS production, which in turn inhibited PMCA activity, thereby causing the slow accumulation of intracellular Ca2+. This increase in [Ca2+]i was in turn able to induce NO release. Likewise, pharmacological blockade of PMCA caused a slow rising elevation in [Ca2+]i that led to robust NO release. AITC was known to elicit ROS-mediated, TRPA1-independent Ca2+ signals, but the underlying signalling machinery was unclear. Thus, this study uncovered for the first time the mechanism by which AITC triggers intracellular Ca2+ signals and Ca2+-dependent functions without involving TRPA1. The study has been carefully conducted, and its conclusions are potentially useful for designing novel strategies to rescue cerebral blood flow in brain disorders. Nevertheless, a number of issues must be addressed before it is suitable for publication: 

TRPA1-mediated Ca2+ signals have long been known to induce neurovascular coupling in brain mice and can be elicited by AITC (PMID: 22928941) or by 4-HNE (PMID: 25564678). The data presented by the authors show that AITC does not activate TRPA1 in hCMEC/D3 cells, but it is unlikely that this channel does not activate work at all in these cells. Is TRPA1 sensitive to 4-HNE in hCMEC/D3 cells?

 Furthermore, are the authors able to check whether AITC does not activate TRPA1-mediated transmembrane currents? For instance, by using the whole-cell patch-clamp technique? Can the authors convincingly rule out that AITC evokes sub-membrane Ca2+ sparklets that are missed by their detection system?

 The authors provide evidence to suggest that AITC-induced ROS production inhibits PMCA activity. Additional experiments to support this conclusion are, however, recommended. Does the exogenous administration of H2O2 inhibits the decay of CPA-evoked intracellular Ca2+ release? Based upon the data presented throughout the manuscript, it should do so.

 Still related to this point, does exogenous administration of H2O2 induce NO release? Again, it is likely that it does.

 It is not clear why RYRs were not inhibited in this study as they are also sensitive to ROS.

 It is not clear how CPA and nigericin, respectively, deplete the ER and lysosomal Ca2+ stores.

Reviewer 2 Report

Dear Authors,

very interesting and well presented manuscript on the regulation of NO release in the cerebrovascular endothelium. I really enjoyed the reading of the manuscript and I only have a few suggestions, questions and reccomendations that I think would further improve the article:

- Page 3, line 102: please define the abbreviation mCRC

- Page 4, lines 152-156: please revise the sentence, there is a repetition and something is missing

- Figure 1A: are the tracing representative or an average of multiple measurements, please specify (this applies to all similar graphs)? Did you try adding the inhibitors after AITC, or performing a washout?

- Please briefly describe the protocols including a washout (e.g.: Figure 3B, 4D)

- Figure 5B: would it be possible/meaningful displaying the results as % decrease from the respective peak value?

- Figure 5G-H: I think they would fit better in Figure 6

Furthermore, it would be interesting to speculate in the discussion what would be the pathophysiologic effect of oxidative stress on blood perfusion/vasodilation.

Minor typos

Round 2

Reviewer 1 Report

The authors have answered all of my questions and the paper has been greatly improved.